# Simple ruthenium-catalyzed reductive amination enables the synthesis of a broad range of primary amines

Thirusangumurugan Senthamarai[1], Kathiravan Murugesan[1], Jacob Schneidewind [1], Narayana V. Kalevaru[1], Wolfgang Baumann[1], Helfried Neumann[1], Paul C.J. Kamer[1], Matthias Beller [1] & Rajenahally V. Jagadeesh [1]

The production of primary benzylic and aliphatic amines, which represent essential feed-stocks and key intermediates for valuable chemicals, life science molecules and materials, is of central importance. Here, we report the synthesis of this class of amines starting from carbonyl compounds and ammonia by Ru-catalyzed reductive amination using $H_2$. Key to success for this synthesis is the use of a simple $RuCl_2(PPh_3)_3$ catalyst that empowers the synthesis of >90 various linear and branched benzylic, heterocyclic, and aliphatic amines under industrially viable and scalable conditions. Applying this catalyst, $-NH_2$ moiety has been introduced in functionalized and structurally diverse compounds, steroid derivatives and pharmaceuticals. Noteworthy, the synthetic utility of this Ru-catalyzed amination protocol has been demonstrated by upscaling the reactions up to 10 gram-scale syntheses. Further-more, in situ NMR studies were performed for the identification of active catalytic species. Based on these studies a mechanism for Ru-catalyzed reductive amination is proposed.

---

[1] Leibniz-Institut für Katalyse e. V. an der Universität Rostock, Albert-Einstein-Str. 29a, 18059 Rostock, Germany. Correspondence and requests for materials should be addressed to M.B. (email: Matthias.Beller@catalysis.de) or to R.V.J. (email: Jagadeesh.Rajenahally@catalysis.de)

The development of efficient catalytic reactions for the selective and sustainable synthesis of amines from readily available and inexpensive starting materials by utilizing abundant and green reagents continues to be an important goal of chemical research[1–6]. In particular, the development of simple and easily accessible catalysts for reductive aminations is highly important because these reactions allow for the cost-efficient production of different kinds of amines[7–30]. Among reductive aminations, the reaction of carbonyl compounds with ammonia in presence of molecular hydrogen to produce primary amines is of central importance and continues to be a major challenge[17–30]. In general, amines are essential chemicals used widely in many research areas and industrial productions related to chemistry, medicine, biology, and material science[1–6]. The majority of existing pharmaceuticals, agrochemicals, biomolecules, and natural products contain amine functionalities, which constitute key structural motifs and play vital roles in their functions[1–6]. Among different kinds of amines, primary benzylic and aliphatic amines constitute valuable fine and bulk chemicals, that serve as versatile feedstocks and key intermediates for advanced chemicals, life science molecules and polymers[1–34]. Regarding their synthesis, catalytic reductive amination of carbonyl compounds (aldehydes and ketones) with ammonia in presence of molecular hydrogen represents a waste-free process to access various linear and branched benzylic and aliphatic amines[17–27]. In addition, catalytic amination of alcohols with ammonia also constitutes a sustainable methodology to produce primary amines[35–38]. Apart from transition metal-catalyzed aminations, the Leuckart–Wallach reaction[39–41] and reduction of oxime ethers with borane[42–44] have also been applied. Noteworthy, selective introduction of primary amine moieties in functionalized compounds by utilizing ammonia constitutes a benign and economic methodology[17–27]. Ammonia, which is produced in >175 million tons per year scale, is considered to be an abundant and green chemical used enormously for the large scale production of urea and other fertilizers as well as various basic chemicals[45–50]. Although ammonia is used extensively for the production of simple molecules, its reactions still encounter common problems such as the requirement of high temperatures or pressures and low selectivity towards the formation of a single desired product[45–50]. Hence, the development of more active and selective catalysts for an effective utilization of ammonia, especially for its insertion in advanced and complex molecules, is highly demanded and challenging.

Reductive amination for the preparation of primary amines, especially in industry, is mainly carried out using heterogeneous catalysts[17–23]. Compared to heterogeneous catalysts, homogeneous catalysis for amination of structurally diverse molecules is less studied and remains challenging[24–27]. Transition metal-catalyzed reactions involving ammonia are often difficult to perform or do not even occur. This problem is mainly due to the deactivation of homogeneous catalysts by the formation of stable Werner-type ammine complexes as well as due to the harsh conditions required for the activation of ammonia. In addition, common problems in reductive aminations, such as over alkylation and reduction to the corresponding alcohols, also affect catalyst viability. In order to utilize ammonia successfully and to overcome these problems, there is a need to develop highly efficient homogeneous catalysts, which is the prime task of this investigation. To date, a few catalysts based on Rh-[24,25] Ir-[25] and Ru-[26,27] complexes were reported for the preparation of primary amines from carbonyl compounds and ammonia using hydrogen. Initially, Beller and co-workers[24] have reported a [Rh(COD)Cl]₂-TPPTS catalyst system for the synthesis of simple primary amines from aldehydes and aqueous ammonia using NH₄OAc as additive. Following this work, Rh[(dppb)(COD)]BF₄ and [Rh(COD) Cl]₂-BINAS catalysts were also applied[25]. Next, [Ir(COD)Cl]₂-BINAS was found to be able to catalyze the amination of a few simple ketones with ammonia[25]. Regarding Ru-catalysts, RuHCl (CO)(PPh₃)₃-xantphos/-dppe in presence of Al(OTf)₃ is known to catalyze the preparation of simple primary amines from ketones[26]. Recently, RuHCl(CO)(PPh₃)₃-(S,S)-f-binaphane[27] in presence of NaPF₆ or NH₄I using NH₃, as well as Ru(OAc)₂-C₃-TunePhos[30] using NH₄OAc have been used for enantioselective reductive amination of ketones to obtain chiral primary amines. These homogeneous catalysts, however, have only been applied in (enantioselective) reductive aminations of simple substrates and have not been used for the preparation of functionalized amines. Despite these advances, the design of simpler yet efficient homogeneous catalysts for the preparation of a broad range of structurally diverse primary amines is highly desired and continues to be an important task from both a research and an industry perspective.

In a lot of cases, homogeneous catalysts applied for challenging reactions and advanced organic synthesis operations are based on sophisticated or synthetically demanding metal complexes and ligands. However, a fundamental and economically important principle is that to achieve a convenient and practical chemical synthesis, the catalyst must be simple, effective and commercially available and/or easily accessible. In this regard, triphenylphosphine (PPh₃)-based metal complexes are found to be expedient and advantageous for catalysis applications, since PPh₃ is a stable and comparatively cheap ligand[51–55]. Among PPh₃-based Ru-complexes, RuCl₂(PPh₃)₃ is considered to be the simplest and least expensive one and is also commercially available. Interestingly, RuCl₂(PPh₃)₃ is known to catalyze a number of organic reactions[56–62]. Herein we demonstrate that RuCl₂(PPh₃)₃ is an efficient and highly selective homogeneous precatalyst for reductive amination, allowing the preparation of a variety of primary amines of industrial importance. By applying this Ru-precatalyst and starting from inexpensive and readily available carbonyl compounds (aldehydes, ketones), ammonia and molecular hydrogen, we undertook the synthesis of functionalized and structurally diverse linear and branched benzylic, heterocyclic, and aliphatic amines including drugs and steroid derivatives. Another objective is to demonstrate up-scaling of the homogeneous amination protocol to gram-scale syntheses. Furthermore, efforts were also made to identify catalytically active species and reaction intermediates by performing kinetic and in situ NMR investigations. Based on these studies, a plausible reaction mechanism is proposed.

## Results

**Selection of catalyst and reaction conditions.** Reductive amination of benzaldehyde (1) to benzylamine (2) with ammonia using molecular hydrogen was chosen as a benchmark reaction. At first, in presence of PPh₃ different metal precursors were tested. As shown in Table 1, the in situ generated Fe-, Co-, Mn-, Ni- and Cu-PPh₃ complexes were not active for the formation of benzylamine (Table 1 entries 1–5). However, in situ generated Ru (II)-PPh₃ complexes showed some activity and produced benzylamine in up to 40% yield (Table 1, entries 6 and 7). After observing this reactivity, we next tested in situ generated Ru-complexes with differently substituted PPh₃-type ligands as well as simple nitrogen ligands (L1–L10). Among these, Ru-catalysts containing either PPh₃ or derivatives with electron donating groups in *para* position showed the highest activity (Table 2; entries 1,4,5). However, none of the tested nitrogen ligands (L7–L10) produced the desired product (Table 2, entries 7–10). Unfortunately, using in situ generated Ru-complexes the yield of benzylamine did not improve beyond 53% (Table 2).

**Table 1 Reductive amination of benzaldehyde: activity of different catalysts**

$$\text{benzaldehyde} + NH_3 \xrightarrow[\text{40 bar } H_2,\ t\text{-amyl alcohol, 130 °C}]{\text{Catalyst (2 mol\%)}} \text{benzylamine-}NH_2$$

| Entry | Metal precursor | L | Yield of benzylamine [%] |
|-------|-----------------|-----|--------------------------|
| 1 | $FeCl_2$ | $PPh_3$ | nd |
| 2 | $CoCl_2 \cdot 6H_2O$ | $PPh_3$ | nd |
| 3 | $MnCl_2$ | $PPh_3$ | nd |
| 4 | $NiCl_2 \cdot 6H_2O$ | $PPh_3$ | nd |
| 5 | $CuCl_2$ | $PPh_3$ | nd |
| 6 | $[RuCl_2(p\text{-cymene})]_2$ | $PPh_3$ | 40 |
| 7 | $[RuCl_2(\text{benzene})]_2$ | $PPh_3$ | 35 |
| 8 | $[RuCl_2(p\text{-cymene})]_2$ | - | nd |
| 9 | - | $PPh_3$ | nd |

Reaction conditions: 0.5 mmol benzaldehyde, 2 mol% metal precursor, 6 mol% $PPh_3$, 5–7 bar $NH_3$, 40 bar $H_2$ 1.5 mL $t$-amyl alcohol, 130 °C, 24 h, GC yields using n-hexadecane as standard
*L* Ligand, *nd* not detected

Next we turned our interest to molecularly defined Ru-complexes. To our delight, the commercially available complexes $RuCl_2(PPh_3)_3$ and $RuCl_2(PPh_3)_4$ showed excellent activity and selectivity for the formation of benzylamine in 92–95% yields (Table 2, entries 11–12). Further, Ru(tris(4-methoxyphenyl)phosphine)$_3$Cl$_2$ and Ru(tris(4-chlorophenyl) phosphine)$_3$Cl$_2$ were also prepared and tested for their reactivity (Table 2; entries 13 and 14). The former displays similar activity compared to $RuCl_2(PPh_3)_3$ (Table 2, entry 13), while the latter was less active (Table 2, entry 14), reflecting the same ligand trend observed in case of in situ generated complexes. In presence of highly active catalysts, we observed 4–7 % of benzyl alcohol (**3**) as the side-product (Table 2, entries 11–13). In case of less-active and or non-active catalysts, undesired side products such as N-benzylidenebenzylamine (**4**) and 2,4,5-triphenyl-2-imidazoline (**5**) were formed (Table 2, entries 1–10). Dibenzylamine (**6**) was not observed under any of these conditions.

**Kinetic investigations**. After having identified $RuCl_2(PPh_3)_3$ as one of the most active precatalysts, we performed kinetic investigations on this system and examined the effect of (a) reaction time, (b) catalyst concentration, (c) hydrogen pressure, (d) reaction temperature, (e) ammonia pressure, and (f) substrate (benzaldehyde) concentration on activity and product distribution (Fig. 1). For reaction time, in Fig. 1a it can be seen that after 5 h, secondary imine **4** is predominantly present (ca. 60%), with only 30% of target product **2**. Over the course of the reaction, **4**, which appears to be an intermediate, is consumed to yield up to 95% **2** after 24 h (for the mechanism of this transformation vide

infra). During the reaction, an increasing amount (up to 4%) of benzyl alcohol is also formed. The cyclic side product **5** can be observed at various reaction times and its amount appears to decrease. This trend, however, is presumably an artifact of the kinetic measurements (see SI). From Fig. 1a it can be concluded that 24 h is an ideal reaction time to obtain maximum yield of **2**. Fig. 1b shows how catalyst loading affects the product distribution. At lower (<2 mol%) loadings, increased amounts of intermediate **4** and side product **5** are obtained, while beyond 2 mol% almost no **4** or **5** along with maximum yield of **2** and some benzyl alcohol **3** were observed. A catalyst loading of 2 mol% is therefore necessary to achieve excellent yield of benzylamine. Similar trends in the product distribution are observed for varied $H_2$ pressure (Fig. 1c) and reaction temperature (Fig. 1d). Thus 40 bar $H_2$ pressure and 130 °C reaction temperature are found to be optimum to suppress the formation of intermediates/side products (**4**/**5**) and to yield maximum amounts of the target product benzylamine. When investigating the effect of ammonia pressure, we found that at less than 5 bar side product **6** (dibenzylamine) is formed in up to 20% (Fig. 1e) yield with a concomitant decrease in the yield of **2**. Hence, a minimum $NH_3$ pressure of 5 bar is required to selectively form the desired product **2** (for mechanistic details vide infra). Further, on increasing the concentration of benzaldehyde (>0.5 mmol) the amount of **5** gradually increases, leading to formation of **5** in up to 80% yield (for 2 mmol benzaldehyde) (Fig. 1e).

**Synthesis of linear primary amines from aldehydes**. Under optimized reaction conditions, we explored the scope of $RuCl_2(PPh_3)_3$-catalyzed reductive amination for the synthesis of

**Table 2 Reductive amination of benzaldehyde using ruthenium catalysts**

| Entry | Ru-precursor/ Defined Ru-catalyst | L | Yield (%) | | | | |
|---|---|---|---|---|---|---|---|
| | | | 2 | 3 | 4 | 5 | 6 |
| 1 [a] | [RuCl$_2$(p-cymene)]$_2$ | L1 | 40 | 2 | 40 | 17 | - |
| 2 [a] | [RuCl$_2$(p-cymene)]$_2$ | L2 | 5 | - | 60 | 33 | - |
| 3 [a] | [RuCl$_2$(p-cymene)]$_2$ | L3 | 2 | - | 25 | 70 | - |
| 4 [a] | [RuCl$_2$(p-cymene)]$_2$ | L4 | 50 | 5 | 20 | 24 | - |
| 5 [a] | [RuCl$_2$(p-cymene)]$_2$ | L5 | 53 | 4 | 16 | 25 | - |
| 6 [a] | [RuCl$_2$(p-cymene)]$_2$ | L6 | 10 | 2 | 42 | 44 | - |
| 7 [a] | [RuCl$_2$(p-cymene)]$_2$ | L7 | - | - | 20 | 76 | - |
| 8 [a] | [RuCl$_2$(p-cymene)]$_2$ | L8 | - | - | 18 | 79 | - |
| 9 [a] | [RuCl$_2$(p-cymene)]$_2$ | L9 | - | - | 25 | 74 | - |
| 10 [a] | [RuCl$_2$(p-cymene)]$_2$ | L10 | - | - | 30 | 68 | - |
| 11 [b] | RuCl$_2$(PPh$_3$)$_3$ | - | 95 | 4 | - | - | - |
| 12 [b] | RuCl$_2$(PPh$_3$)$_4$ | - | 92 | 7 | - | - | - |
| 13 [b] | RuCl$_2$(tris(4-methoxyphenyl)phosphine)$_3$ | - | 95 | 4 | - | - | - |
| 14 [b] | RuCl$_2$(tris(4-chlorophenyl)phosphine)$_3$ | - | 50 | - | 49 | - | - |

Reaction conditions: [a]0.5 mmol benzaldehyde, 1 mol% [RuCl$_2$(p-cymene)]$_2$ (2 mol% with respect to the monomer), 6 mol% ligand, 5-7 bar NH$_3$, 40 bar H$_2$ 1.5 mL $t$-amyl alcohol, 130 °C, 24 h, GC yields using n-hexadecane as standard. [b]Same as 'a' but using 2 mol% defined catalyst.

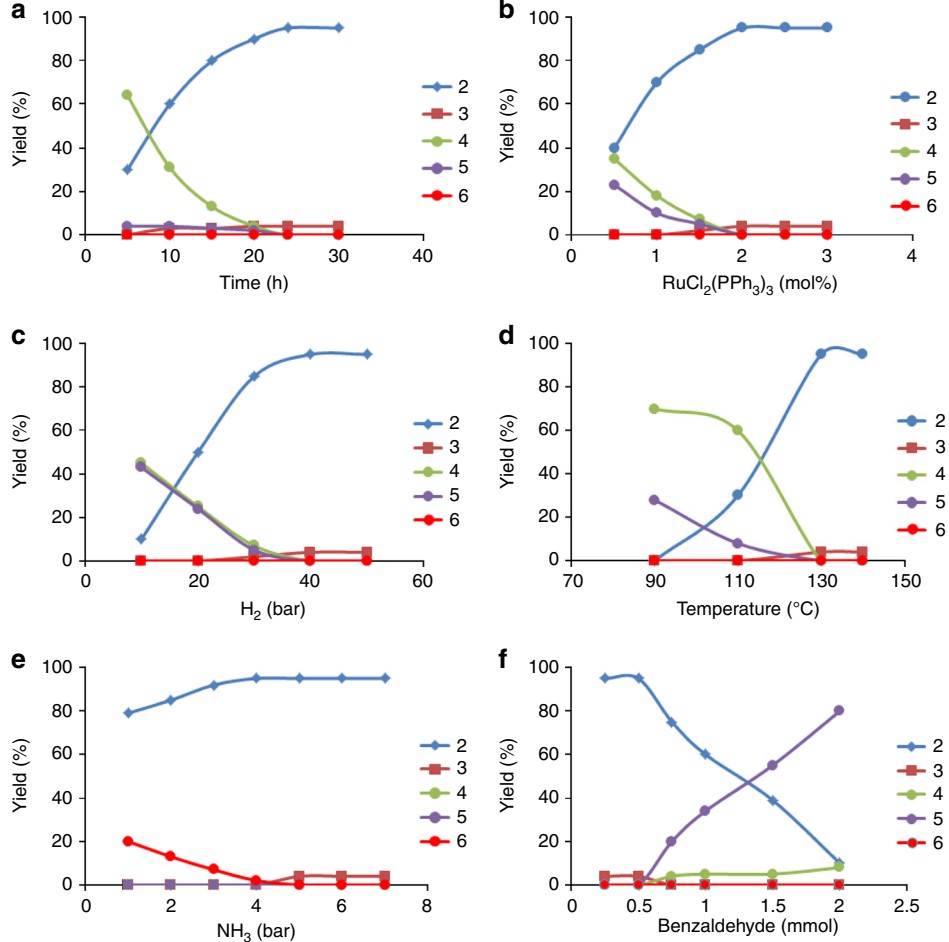

**Fig. 1** Kinetic investigations on the Ru-catalyzed reductive amination of benzaldehyde. **a** Yield vs reaction time, **b** yield vs concentration of $RuCl_2(PPh_3)_3$, **c** yield vs pressure of $H_2$, **d** yield vs temperature, **e** yield vs pressure of $NH_3$, **f** yield vs concentration of benzaldehyde. 2 = Yield of benzylamine; 3 = yield of benzyl alcohol; 4 = yield of N-benzylidenebenzylamine; 5 = yield of 2,4,5-triphenyl-4,5-dihydro-1H-imidazole; 6 = yield of dibenzylamine. Reaction conditions: For Fig. 1a: 0.5 mmol benzaldehyde, 2 mol% $RuCl_2(PPh_3)_3$, 5–7 bar $NH_3$, 40 bar $H_2$, 1.5 mL t-amyl alcohol, 130 °C, 5–30 h; for Fig. 1b: 0.5 mmol benzaldehyde, 0.5–3 mol% $RuCl_2(PPh_3)_3$, 5–7 bar $NH_3$, 40 bar $H_2$, 1.5 mL t-amyl alcohol, 130 °C, 24 h; for Fig. 1c: 0.5 mmol benzaldehyde, 2 mol% $RuCl_2(PPh_3)_3$, 5–7 bar $NH_3$, 10–50 bar $H_2$, 1.5 mL t-amyl alcohol, 130 °C, 24 h; for Fig. 1d: 0.5 mmol benzaldehyde, 2 mol% $RuCl_2(PPh_3)_3$, 5–7 bar $NH_3$, 40 bar $H_2$, 1.5 mL t-amyl alcohol, 90–140 °C, 24 h. for Fig. 1e: 0.5 mmol benzaldehyde, 2 mol% $RuCl_2(PPh_3)_3$, 1–7 bar $NH_3$, 40 bar $H_2$, 1.5 mL t-amyl alcohol, 130 °C, 24 h; for Fig. 1f: 0.25–2 mmol benzaldehyde, 2 mol% $RuCl_2(PPh_3)_3$, 5–7 bar $NH_3$, 40 bar $H_2$, 1.5 mL t-amyl alcohol, 130 °C, 24 h. Yields were determined by GC using n-hexadecane as standard

various primary amines. As shown in Fig. 2, industrially relevant and structurally diverse benzylic, heterocyclic, and aliphatic aldehydes underwent reductive amination and offered linear primary amines in good to excellent yields. Simple as well as sterically hindered benzaldehydes were selectively converted to their corresponding benzyl amines in up to 95% yield (Fig. 2; products **2** and **7–13**). In order to apply this amination methodology for organic synthesis and drug discovery, achieving a high degree of chemoselectivity is important. In this regard, we conducted the reaction of sensitive halogenated and functionalized benzaldehydes. Delightfully, halogen-substituted benzaldehydes, including more sensitive iodo-substituted compounds, selectively underwent reductive amination without any significant dehalogenation (Fig. 2; products **14–21**). Gratifyingly, various functional groups such as ethers, thio-ethers, carboxylic acid-esters and boronic acid-esters, amides and challenging C–C triple bonds were all well-tolerated without being reduced (Fig. 2; products **22–40**). In all these cases, the aldehyde group was selectively aminated to produce functionalized amines in up to 88% yield.

Heterocycles are regarded as highly valuable compounds and these motifs serve as integral parts of a large number of life science molecules and natural products. Thus, the preparation of heterocyclic primary amines is routinely needed en route to the production of pharmaceutically and agriculturally valuable products. Consequently, a series of different heterocyclic amines were synthesized (Fig. 2; products **41–49**). The primary amines of pyridine, methylenedioxybenzene and benzodioxane, furan and thiophene were obtained in 87–92% yields.

Success in the amination of aromatic and heterocyclic aldehydes prompted us to validate this catalyst also for aliphatic substrates. Commonly, amination of aliphatic aldehydes is more challenging and most reported catalysts exhibit lower reactivity towards these substrates. In addition, the reaction of aliphatic aldehydes is often troubled by the formation of unwanted aldol reaction products. In spite of these problems, the $RuCl_2(PPh_3)_3$ precatalyst is found to be highly active and selective for the preparation of aliphatic primary linear amines too (Fig. 2). Accordingly, various primary araliphatic and aliphatic linear amines including allylic ones (products **56** and **57**) were obtained in up to 92% yield. Importantly, phenylethylamines (products **50** and **51**), which function as monoaminergic neuromodulators and neurotransmitters in the human CNS, have been prepared in up to 90% yield.

**Fig. 2** Ru-catalyzed synthesis of linear primary benzylic, heterocyclic, and aliphatic amines. [a]Reaction conditions: [a]0.5 mmol aldehyde, 2 mol% RuCl$_2$(PPh$_3$)$_3$, 5–7 bar NH$_3$, 40 bar H$_2$ 1.5 mL $t$-amyl alcohol, 130 ºC, 24 h, isolated yields. [b]GC yields using n-hexadecane as standard. [c]same as 'a' for 30 h. Isolated as free amines and converted to hydrochloride salts. Corresponding hydrochloride salts were subjected to NMR analysis

**Fig. 3** RuCl$_2$(PPh$_3$)$_3$-catalyzed synthesis of branched primary amines from ketones. [a]Reaction conditions: 0.5 mmol ketone, 3 mol% RuCl$_2$(PPh$_3$)$_3$, 5–7 bar NH$_3$, 40 bar H$_2$ 1.5 mL t-amyl alcohol, 130 °C, 24 h, isolated yields. [b]GC yields using n-hexadecane as standard. [c]Same as 'a' with 1 mol% catalyst. [d]Same as 'a' for 30 h. [e]Diastereomeric ratio. Isolated as free amines and converted to hydrochloride salts. Corresponding hydrochloride salts were subjected to NMR analysis

**Synthesis of branched primary amines from ketones**. After having successfully performed the reductive amination of aldehydes, we were interested in the general applicability of this ruthenium precatalyst for the synthesis of branched primary amines starting from different ketones (Fig. 3). Compared to aldehydes, the reaction of ketones with ammonia to form primary amines is more difficult. Remarkably, the RuCl$_2$(PPh$_3$)$_3$ precatalyst is also active towards aromatic ketones (Fig. 3). Further, the applicability of this catalyst system was also explored for aliphatic ketones. Here, the aliphatic branched primary amines were obtained in up to 94% yield (Fig. 3).

**Applications to life science molecules**. To showcase the valuable applications of this amination protocol, we carried out the preparation of existing drugs as well as the introduction of –NH$_2$ moieties into drugs and complex molecules. The amination of important drugs such as Nabumetone, Pentoxifylline, and Azaperone (Fig. 4; products **92–94**) as well as steroid derivatives has been demonstrated (Fig. 4; products **95–97**). Such an insertion of amino groups into life science molecules represents a resourceful technique for further functionalization and modulation of their activities, which is highly useful in drug discovery.

**Upscaling for the preparation of amines on gram-scale**. In order to show practical utility and to demonstrate potential for

implementation in industrial production, the upscaling of synthetic methodologies is very important. Especially in homogeneous catalysis upscaling is a challenging task. Therefore, to demonstrate the applicability of this homogeneous catalytic amination protocol, we performed gram-scale synthesis of six selected amines. As shown in (see Supplementary Figure 1), 2–10 g of four aldehydes and two ketones were successfully aminated to yield their corresponding primary amines in more or less similar yields to those of 50–100 mg scale reactions.

We were interested to compare our methodology to an established amination protocol. The Leuckart-Wallach reaction is a prime example, finding application also on an industrial scale[39–41]. We therefore subjected 15 aldehydes and ketones, which have been studied in this work, to Leuckart-Wallach reaction conditions[39–41] to prepare the corresponding primary amines. As shown in Supplementary Table 1, the reaction worked well for simple aldehydes/ketones and gave 53–75% of corresponding primary amines (Supplementary Table 1; entries 1–3). For a majority of substituted and structurally diverse as well as heterocyclic aldehydes and ketones it gave poor yields (5–15%) (Supplementary Table 1; entries 4–9). The Leuckart-Wallach reaction failed to yield the desired primary amine for sensitive substrates (e.g., TMS or halogen containing) as well as some (hetero)cyclic and steroid derivatives (Supplementary Table 1; entries 10–15). A majority of the sensitive functional groups were

**89: 85%**
**(2-Methoxyamphetamine)**

**90: 92%**
**(4-Hydroxyamphetamine)**

**91: 82%**
**(4-Hydroxy-3-methoxyamphetamine)**

**92: 90%**
**(Nabumetone-NH₂)**

**93: 76%**
**(Pentoxifylline-NH₂)**

**94: 80%**
**(Azaperone-NH₂)**

**95: 93% (92:8)[b]**
**(Androstanolone-NH₂)**

**96: 79% (52:48)[b]**
**(Estrone-NH₂)**

**97: 92% (90:10)[b]**
**(Testosterone propionate-NH₂)**

**Fig. 4** Synthesis of drugs and amination of complex molecules. [a]Reaction conditions: 0.5 mmol ketone, 3 mol% RuCl₂(PPh₃)₃, 5–7 bar NH₃, 40 bar H₂ 1.5 mL t-amyl alcohol, 130 °C, 24 h, isolated yields. Isolated as free amines and converted to hydrochloride salts. Corresponding hydrochloride salts were subjected to NMR analysis. [b]Diastereomeric ratio

**Fig. 5** Ru-catalyzed reductive amination of carbonyl compounds with NH₃ using H₂. **a** Noncatalytic condensation reaction; **b** catalytic hydrogenation reaction

not tolerated. Gratifyingly, for all these substrates, the RuCl₂(PPh₃)₃ precatalyst using ammonia and hydrogen worked well and produced the corresponding primary amines in 72–93% yields. These results clearly reveal that catalytic reductive amination using RuCl₂(PPh₃)₃ is more generally applicable for the preparation of primary amines compared to the traditional Leuckart-Wallach reaction.

## Discussion

A general reaction pathway for the catalytic reductive amination of carbonyl compounds is shown in Fig. 5. Initially, the carbonyl compound undergoes condensation with ammonia to form the corresponding primary imine. Subsequently, the intermediate imine is hydrogenated to give the primary amine. The hydrogenation step is catalyzed by a catalytic species derived from the precatalyst, RuCl₂(PPh₃)₃.

We were interested to gain mechanistic insight into the hydrogenation step and to determine the nature of the active catalyst species. For this purpose, we studied the interaction of RuCl₂(PPh₃)₃ with hydrogen using in situ NMR in a model system consisting solely of the ruthenium precatalyst, methanol and $C_6D_6$. Figure 6 depicts the hydride region of the obtained $^1H$ NMR spectra. Initially, even in the absence of $H_2$, a quartet at $\delta_H = -17.6$ ppm is observed along with a broad singlet at $\delta_P = 55$ ppm in the $^{31}P\{^1H\}$ NMR spectrum (see SI). We assign these signals to [RuHCl(PPh₃)₃][63], which is likely formed in small amounts via methanol oxidation. In presence of $H_2$ (1.5 bar) at room temperature, the quartet corresponding to [RuHCl(PPh₃)₃] broadens[63] and two new hydride signals appear: a broad singlet at $\delta_H = -12.5$ ppm and a triplet of triplets at $\delta_H = -10.9$ ppm. Using $^1H$-$^{31}P$ HMBC NMR (see Supplementary Figures 6–13) we

were able to assign the hydride triplet of triplets to two multiplets in the $^{31}P\{^1H\}$ NMR spectrum (at $\delta_P = 34.8$ ppm and $\delta_P = 58.4$ ppm; see SI), which is consistent with the structure of [Ru (H)₂(PPh₃)₄][64]. We tentatively assign the broad singlet at $\delta_H = -12.5$ ppm to [Ru(H)₂(PPh₃)₃], which is corroborated by the appearance of a broad signal at $\delta_P = 58$ ppm in the $^{31}P\{^1H\}$ NMR spectrum[65]. [Ru(H)₂(PPh₃)₃] would be in equilibrium with [Ru (H)₂(PPh₃)₄] via association/dissociation of a PPh₃ ligand. After 2.5 h at room temperature a new triplet at $\delta_H = -9.4$ ppm appears in the hydride region, which further increases in intensity upon heating to 60 °C. Using $^1H$-$^{31}P$ HMBC NMR we could assign this hydride signal to a singlet in the $^{31}P\{^1H\}$ NMR spectrum at $\delta_P = 50.4$ ppm (see Supplementary Figures 6–13). After 1.5 h at 60 °C, it is the dominant species in the hydride region and in the $^{31}P\{^1H\}$ NMR spectrum. The triplet hydride splitting (37 Hz), which collapses to a singlet in the $^1H\{^{31}P\}$ NMR spectrum (see Supplementary Figures 6–13), indicates the presence of just two equivalent PPh₃ ligands. When the $^{31}P\{^1H\}$ NMR experiment is decoupled with reduced power (only aromatic protons are decoupled) the singlet at $\delta_P = 50.4$ ppm splits into a doublet (see Supplementary Figures 6–13)), indicating a monohydride structure. Although this species appears to have only two PPh₃ and one hydride ligand, its accumulation indicates high stability under experimental conditions, suggesting the presence of other stabilizing ligands (such as CO). Since [Ru (H)₂(PPh₃)₃] is known to decarbonylate methanol[66] and due to similar spectral characteristics compared to [RuHCl(CO) (PPh₃)₂(pyrazine)][67] we tentatively assign this species to the carbonyl-containing complex [RuHCl(CO)(PPh₃)₂(Y)] (with Y possibly being a solvent molecule) formed via methanol decarbonylation.

An overview of the proposed transformation of RuCl₂(PPh₃)₃ in our model system is provided in Fig. 7: RuCl₂(PPh₃)₃ undergoes a stepwise reaction with $H_2$ to form [RuHCl(PPh₃)₃] (which is also generated by the reaction with methanol) and Ru (H)₂(PPh₃)₃, which is in equilibrium with Ru(H)₂(PPh₃)₄. Ru (H)₂(PPh₃)₃ can further react via alcohol decarbonylation to form the carbonyl-containing complex [RuHCl(CO)(PPh₃)₂(Y)]. While methanol is not present under our reaction conditions for reductive amination, it is known that RuCl₂(PPh₃)₃ can also enable the decarbonylation of benzyl alcohols and aldehydes[65], which constitute a majority of our substrates.

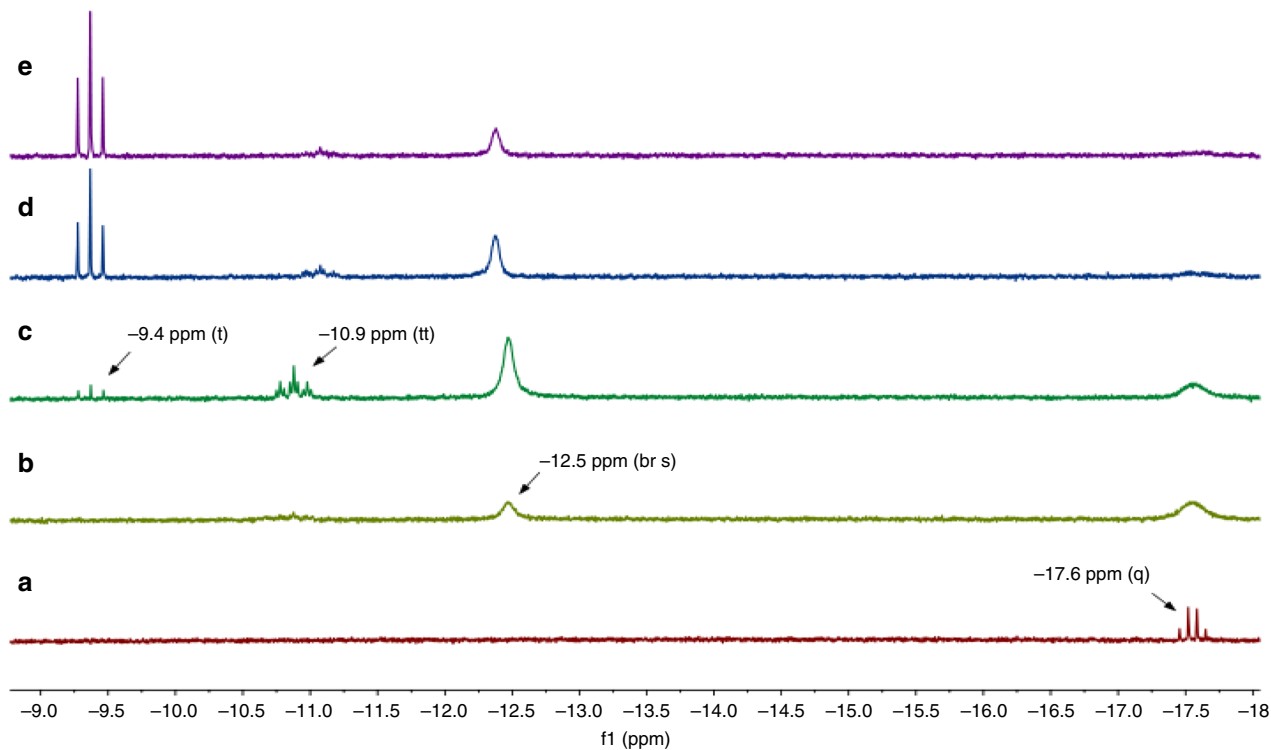

**Fig. 6** Hydride region of $^1H$ NMR spectra of $RuCl_2(PPh_3)_3$ in $C_6D_6$/methanol. **a** RT, argon atmosphere; **b** RT, $H_2$ atmosphere (1.5 bar), 10 min; **c** RT, $H_2$ atmosphere (1.5 bar), 2.5 h; **d** 60 °C, $H_2$ atmosphere (1.5 bar), 30 min; **e** 60 °C, $H_2$ atmosphere (1.5 bar), 1.5 h

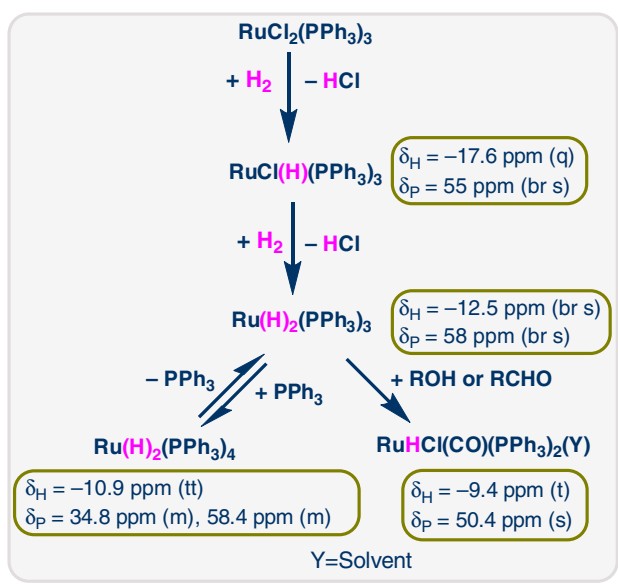

**Fig. 7** Generation of different species from $RuCl_2(PPh_3)_3$ in presence of hydrogen. $RuCl_2(PPh_3)_2$ = precatalyst; $Ru(H)Cl(PPh_3)_3$ and $Ru(H)_2(PPh_3)_3$ = active catalytic species

Also, a number of previously reported ruthenium systems for reductive amination as well as alcohol amination are based on carbonyl-containing (pre-)catalysts[26,27,35–37]. Therefore, the question arises whether our active catalyst contains a carbonyl ligand or if it conforms to a previously proposed $[RuHX(PPh_3)_3]$ structure ($X = H^-$ or $Cl^-$)[68,69]. To answer this question, we have compared the catalytic performance of $[RuHCl(PPh_3)_3]$ and $[RuHCl(CO)(PPh_3)_3]$ for benzaldehyde amination under

standard reaction conditions. Interestingly, $[RuHCl(PPh_3)_3]$ performs similarly to $RuCl_2(PPh_3)_3$ (88% benzylamine, 4% benzyl alcohol, 7% dibenzylamine; Supplementary Figure 2). This confirms that $[RuHCl(PPh_3)_3]$ is part of the transformation cascade (as observed in our model system) which also includes the active catalyst. However, $[RuHCl(CO)(PPh_3)_3]$ showed poor selectivity under our reaction conditions (3% benzylamine, 5% benzyl alcohol, 90% N-benzylidenebenzylamine; Supplementary Figure 3). In addition, a reported $Ru_3(CO)_{12}$/CataCxiumPCy catalytic system, which was used in the amination of alcohols with ammonia[36], was also tested for the reductive amination of cyclohexanone (Supplementary Figure 3). Similarly, this catalyst also showed poor selectivity, yielding only 10% of cyclohexylamine. Therefore, carbonyl-containing complexes are likely not the active species under our reaction conditions. Rather, due to their decreased selectivity, they constitute a possible deactivation pathway for $RuCl_2(PPh_3)_3$ catalyzed reductive amination. This difference between our observations and previously reported carbonyl-containing ruthenium amination catalysts is attributed to the ligand: carbonyl-containing Ru(II) catalysts typically require bidentate[26] or tridentate[35] ligands. Control experiments have shown significantly decreased yield in the absence of those additional ligands[26]. In contrast, the carbonyl-free catalyst type $[RuHX(PPh_3)_3]$ appears to be sufficiently active and selective with only $PPh_3$-derived ligands.

After clarifying the pathway for catalyst activation we were interested to investigate the reaction cascade starting from the aldehyde/ketone and ammonia using the benzaldehyde benchmark system. The starting materials can undergo a condensation to form primary imine **A**. Intermediate **A**, however, was never detected in the reaction mixture, presumably due to its high reactivity. Instead (as can be seen in Fig. 1a), secondary imine **4** was determined to be the major intermediate. **4** is formed via condensation of the product **2** with either the starting aldehyde/ketone (releasing water) or via condensation of **2** and **A** (releasing

**Fig. 8** Proposed reaction mechanism for the RuCl$_2$(PPh$_3$)$_3$-catalyzed reductive amination. **A** Unstable primary imine; **A′** stable secondary imine. **5** = 2,4,5-triphenyl-2-imidazoline

NH$_3$). We next reacted isolated **4** under our standard reaction conditions (40 bar H$_2$, 5–7 bar NH$_3$, 24 h, RuCl$_2$(PPh$_3$)$_3$) and found almost quantitative conversion to **2** (Supplementary Figure 4). In contrast, when **4** was reacted in the absence of ammonia, 98% dibenzylamine was obtained after 24 h (Supplementary Figure 4). These results show that **4**, when exposed to NH$_3$, is in an equilibrium with **A** + **2**[70]. While the catalyst is able to hydrogenate both **A** and **4**, **A** is hydrogenated preferentially and is replenished by the equilibrium with **4**. If ammonia is absent or only present in low concentrations (see Fig. 1e), the formation of **A** + **2** from **4** is suppressed, leading to hydrogenation of **4** (yielding dibenzlamine). Furthermore, due to rapid hydrogenation under optimized conditions, the stationary concentration of **A** is low, precluding side reactions of this reactive intermediate. When the hydrogenation does not proceed quickly, however, accumulation and side reactions involving **A** can likely occur. Correspondingly, when a mixture of **4** and benzaldehyde was reacted under standard conditions but without H$_2$, 20% of the cyclic side-product **5** was obtained (the rest being unreacted starting material, see Supplementary Figure 5). Williams et al.[71] and Corey et al.[72] have reported that **A** can trimerize to form **99**, which can subsequently undergo thermal cyclization to form **5** (Fig. 8). This reaction of accumulated **A** likely explains the formation of large amounts of **5** in less active catalyst systems (see Table 1, entries 1–10; Fig. 1).

Based on these observations we propose the following mechanism (Fig. 8): Reaction of a carbonyl compound with NH$_3$ yields primary imine **A**, which can be in an equilibrium with secondary imine **A′** via condensation with the product amine. The precatalyst RuCl$_2$(PPh$_3$)$_3$ is activated by H$_2$ to form the the active catalyst species [RuHX(PPh$_3$)$_3$] (X being either H$^-$ or Cl$^-$). This active catalytic species selectively reacts with the primary imine to initially form a substrate complex (**I**). Substrate coordination is followed by hydride insertion (**II**), generating a Ru-amide complex. Coordination of H$_2$ (**III**) followed by hydrogenolysis releases the primary amine as the final product with regeneration of the catalytic species (**IV**).

In conclusion, we demonstrated that using a simple RuCl$_2$(PPh$_3$)$_3$ catalyst, the challenging reductive amination of carbonyl compounds using ammonia and molecular hydrogen for the selective synthesis of a variety of primary amines is possible. Applying this Ru-based reductive amination, starting from inexpensive aldehydes and ketones, functionalized and structurally diverse linear and branched primary amines have been synthesized under industrially viable and scalable conditions. In general, achieving a high degree of chemoselectivity in amination/hydrogenation reactions is a challenging task. In this regard our simple Ru-based methodology represents a unique example in homogeneous catalysis for the reductive amination of functionalized and challenging molecules. We have also shown the possibility of scaling this amination protocol up to 10 g without any loss in either activity or selectivity. The application of this approach is also extended to the synthesis and amination of various drug molecules and steroid derivatives. In situ NMR investigations provided clear hints on the formation of Ru-hydride species, which have been elucidated to be the active catalytic species in this RuCl$_2$(PPh$_3$)$_3$-catalyzed reductive amination. With the help of these investigations, an appropriate reaction mechanism has been proposed.

## Methods

**General considerations**. All carbonyl compounds (aldehydes and ketones), Ru-precursors and complexes and ligands, were obtained commercially. All catalytic experiments were carried out in 300, 100, and 500 mL autoclaves (PARR Instrument Company). In order to avoid unspecific reactions, all catalytic reactions were carried out either in glass vials, which were placed inside the autoclave, or glass/Teflon vessel fitted autoclaves. GC and GC-MS were recorded on a Agilent 6890N instrument. GC conversion and yields were determined by GC-FID, HP6890 with FID detector, column HP530 m × 250 mm × 0.25 μm. $^1$H, $^{13}$C NMR data were recorded on a Bruker AV 300 and Bruker AV 400 spectrometers using DMSO-d$_6$, CD$_3$OD or C$_6$D$_6$ as solvents. Ru(tris(4-methoxyphenyl)phosphine)$_3$Cl$_2$ and Ru(tris(4-chlorophenyl)phosphine)$_3$Cl$_2$ were prepared according to the reported procedure[73].

**Reductive amination of carbonyl compounds with ammonia**. The 8 mL dried glass vial was charged with a magnetic stirring bar and 0.5 mmol of corresponding carbonyl compound (aldehyde or ketone). Then 1.5 mL t-amyl alcohol as solvent and 2–3 mol% RuCl$_2$(PPh$_3$)$_3$ catalysts (2 mol% in case of aldehydes and 3 mol% in case of ketones) were added. The glass vial was fitted with a septum, cap and needle, and placed into a 300 mL autoclave (eight vials with different substrates at a time). The autoclave was flushed with hydrogen twice at 40 bar pressure and then it was pressurized with 5–7 bar ammonia gas and 40 bar hydrogen. The autoclave was

placed into an aluminum block preheated at 140 °C (placed inside 30 min before counting the reaction time in order to attain the reaction temperature) and the reactions were stirred for the required time. During the reaction the inside temperature of the autoclave was measured to be 130 °C and this temperature was used as the reaction temperature. After completion of the reactions, the autoclave was cooled to room temperature. The remaining ammonia and hydrogen were discharged and the vials containing reaction products were removed from the autoclave. The reaction products were analyzed by GC-MS and the corresponding primary amines were purified by column chromatography (silica; n-hexane-ethyl acetate mixture). The resulting amines were converted to their respective hydrochloride salt and characterized by NMR. For conversion into the hydrochloride salt, 1–2 mL methanolic HCl or dioxane HCl (1.5 M HCl in methanol or 4 N HCl in dioxane) was added to the ether solution of the respective amine and stirred at room temperature for 4–5 h. Then, the solvent was removed and the resulting hydrochloride salt of the amine was dried under high vacuum. For determining the yields by GC for selected amines, after completion of the reaction n-hexadecane (100 μL) as standard was added to the reaction vials and the reaction products were diluted with ethyl acetate followed by filtration using a plug of silica and then analyzed by GC.

**General procedure for the gram scale reactions**. The Teflon or glass fitted 300 (5–10 g) or 500 mL (20 g) (in case 5–20 g) or 100 mL (in case of 2–2.5 g) autoclave was charged with a magnetic stirring bar and the corresponding carbonyl compound (2–20 g). Then 25–150 mL t-amyl alcohol was added. Subsequently, RuCl$_2$(PPh$_3$)$_3$ (amount of catalysts equivalent to 2–3 mol%) was added. The autoclave was flushed with hydrogen twice at 40 bar pressure and then it was pressurized with 5–7 bar ammonia gas and 40 bar hydrogen. The autoclave was placed into an aluminum block preheated to 140 °C (placed 30 min before counting the reaction time in order to attain reaction temperature) and the reaction was stirred for the required time. During the reaction the inside temperature of the autoclave was measured to be 130 °C and this temperature was used as the reaction temperature. After completion of the reaction, the autoclave was cooled to room temperature. The remaining ammonia and hydrogen were discharged and the reaction products were removed from the autoclave. The reaction products were analyzed by GC-MS and the corresponding primary amines were purified by column chromatography (silica; n-hexane-ethyl acetate mixture). The resulting amines were converted to their respective hydrochloride salt and characterized by NMR.

**Procedure for the in situ NMR studies**. The in situ observation of the Ru-hydrides was performed under hydrogen saturation conditions in a 5 mm glass NMR tube, equipped with a PTFE gas inlet hose and a circulation unit which produces a continuous gas flow through the solution[74,75]. The brown solution of the precursor complex RuCl$_2$(PPh$_3$)$_3$ (50 mg) in 0.5 mL methanol/0.5 mL benzene-d6 was transferred to the NMR tube under Ar. After assembling the device under inert gas and characterizing the solution by its $^1$H and $^{31}$P NMR spectra, the system was filled with neat hydrogen (absolute pressure 1.5 bar). A gas flow of 1 mL min$^{-1}$ was used to saturate the solution. $^1$H and $^{31}$P NMR spectra were taken at regular intervals to monitor the reaction progress. Changes were immediately observable as shown in Fig. 2. After three hours, the temperature was raised and kept at about 60 °C for another three hours to complete the reaction. No further changes were detected thereafter. The color of the solution was changed to brick-red at the end of the experiment. Note that maintaining a continuous gas flow till the very end was not possible, because black particles of precipitating metallic Ru were clogging the tubing.

## Data availability

All data are available from the authors upon reasonable request.

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

## Acknowledgements

We gratefully acknowledge the European Research Council (EU project 670986-NoNa-Cat) and the State of Mecklenburg-Vorpommern for financial and general support. Financial support by Fonds der Chemischen Industrie (Kekulé-Stipendium n. 102151) for J.S. is also acknowledged. We thank the analytical staff of the Leibniz-Institute for Catalysis, Rostock for their excellent service.

## Author contributions

R.V.J., T.S., K.M., M.B., and N.V.K. planned and developed the project. T.S., K.M., R.V.J., J.S., N.V.K., H.N., and P.C.J.K. designed the experiments. T.S. and K.M. performed catalytic experiments. W.B. and J.S. performed in situ NMR experiments. R.V.J., J.S., N.V.K., T.S., K.M., P.C.J.K., and M.B. wrote the paper. R.V.J. and M.B. supervised the project.

## Additional information

**Competing interests:** The authors declare no competing interests.

