## [Peer Review File · Nature Communications]

Reviewer #1 (Remarks to the Author):

The research group of Beller and Jagadeesh have developed a synthetic method for the primary amines from carbonyl compounds using ammonia, molecular hydrogen and simple ruthenium precursor as a catalyst. Beller and other research group have reported Rh, Ir and including Ru based catalytic system for the similar transformation. However, the strength of this manuscript is widespread applicability and ardent display of substrate scope. In fact, such extensive investigations on functionalized and structurally diverse aldehydes and ketones leading to the highly useful primary amines are performed for the first time. Impressive scalability and standard mechanistic investigations have also been reported. This manuscript can certainly influence others in the field to develop strong substrate scope. This reviewer recommends publication of this work in Nature Communications after addressing following comments.

- 1) Lines 77-79: redundant
- 2) Lines 113: "design of catalyst" has to be changed as selection of catalyst (no new catalyst is designed for this work).
- 3) PPh₃
- 4) Lines 117 and 125: typos
- 5) Table 2: desired conversion and yields have been attained in first two entries with no added ligands. Then, what is the need to have extensive ligands screening with poorly active catalyst in entries 3-12?
- 6) Figure 1: kinetic experiments clearly reveal that the homo-coupled imines as major intermediate (ca 60%). Thus, under the experimental conditions one would expect the presence of the corresponding hydrogenation product, secondary amine. Still no such product is observed (Table 2). Since the major intermediacy of internal imine is established in kinetics, a reaction pathway for the conversion of internal imines to primary amines need to be considered in mechanistic proposal.
- 7) Reconsider "for measuring NMR".
- 8) Fig 2, compound 48: drawing error
- 9) Fig 6: diastereomeric ration shown for compound 82 but not 85.
- 10) RuCl₂(PPh₃) is described as both catalyst and pre-catalyst.
- 11) Lines 326-332: ³¹P {¹H} NMR Spectrum

- 12) Fig 12: as the intermediacy of $[\text{RuHCl}(\text{PPh}_3)_3]$ is established and it is readily available, reaction of it with independently prepared imine may allow prediction of more relevant intermediates.
- 13) Line 418: vials not vails
- 14) Formation of metallic particles observed in NMR investigations. What about their involvement in reactions?
- 15) Impressive, in fact more attractive synthetic method for the primary amines is reported directly from primary (by Milstein group: Gunanathan, C.; Milstein, D. *Angew. Chem., Int. Ed.* 2008, 47, 8661) and secondary alcohols (by Beller group and others: Imm, S.; Bañ, S.; Neubert, L.; Neumann, H.; Beller, M. *Angew. Chem., Int. Ed.* 2010, 49, 8126 and Pinggen, D.; Müller, C.; Vogt, D. *Angew. Chem., Int. Ed.* 2010, 49, 8130) using ruthenium catalysts. As the carbonyl intermediacy is proposed in all such work, discussion of them in this manuscript is must (I am surprised by this omission).

Chidambaram Gunanathan

Reviewer #2 (Remarks to the Author):

Summary of major claims of the paper:

In this work, Jagadeesh, Beller, and co-workers show how a simple ruthenium based catalyst can effect the reductive amination of a variety of aldehydes and ketones. A number of first-row transition metals were screened, which showed the unique effectiveness of ruthenium. A screen of ligands showed triphenylphosphine was an effective ligand. A large substrate scope is presented. Aldehydes, including some very sterically hindered variants, are aminated in excellent yield. Halogens, a pinacolboron group, and a TMS protected alkyne are tolerated. Impressively, aldehydes with conjugated alkenes can be subject to reductive amination without concomitant reduction. At 3 mol% loading, ketones, including some fairly sterically hindered examples, can be aminated in good yield. A number of pharmaceutically relevant molecules were also aminated under these conditions. Seven examples were conducted on multigram scale, demonstrating the utility of this method. A mechanistic study suggests the intermediacy of ruthenium hydrides, and also suggests that

ruthenium carbonyl complexes, which may be formed under the reaction conditions, are probably not catalytically relevant.

Reviewer's opinion on merit of the work:

While this is a nice piece of work, on a timely topic, I am not sure that as it stands that this paper will strongly influence thinking in the field. This work references two recently published papers, by the Schaub and Zhang groups (references 27 and 30), which conduct asymmetric reductive amination of ketones with hydrogen and ammonia or ammonium salts and chiral ruthenium complexes. These papers, which were published while this large body of work was presumably in progress, allow the asymmetric formation of amines with a lower loading of ruthenium than the present work. Nevertheless, this current work may have merit, since both of these papers use exotic and expensive ligands, whereas the current work uses simple and inexpensive ligands. In many cases, the resolution of racemic amines (through enzymatic resolution, or diastereomeric crystallization) is still competitive with their asymmetric synthesis, accordingly, a new synthesis of racemic amines, such as this work does have value.

The high pressures and temperatures required in this work probably make it more amenable to industrial use than small-scale transformations. Racemic amines are commonly prepared by the Leuckart-Wallach reaction, which does not require the use of transition metals, and is economical and scalable. Would the authors be able to present an analysis that the large scale reactions presented in this work are competitive with Leuckart-Wallach chemistry? Can the loading of ruthenium be lowered? Are the amounts of ruthenium residue in the product below acceptable thresholds for pharmaceutical APIs?

While this work is presented as a way to prepare diverse families of amines on small scale, the formation and reduction of oxime ethers with borane (which should be referenced) is a very reliable way to prepare primary amines on small scale without the use of an autoclave.

Specific questions:

Table 2 is somewhat uninformative. Entries 1 and 2, involving the use of preformed "molecularly defined" catalysts based on triphenylphosphine provided the best performance. All other phosphines were employed to generate the catalyst in-situ, with a ruthenium cymene complex as

the ruthenium source. These in-situ generated catalysts all had inferior performance to the preformed catalysts, even in the case of triphenylphosphine. Accordingly, I feel in a ligand comparison, it would have made more sense to use preformed catalysts with some other phosphine ligands as well.

Kinetic analysis shows the presence of side products. I would like to see further commentary and analysis regarding the formation and disappearance of compound 5. Is this compound, which must arise from a series of redox reactions, including a C-C bond formation, turned into product, or polymeric material after time?

Reviewer #3 (Remarks to the Author):

In the current manuscript, Beller and Jagadeesh describe a ruthenium-catalyzed reductive amination. As such this process is not new, but it is more general. Previous reports from the Beller group (using rhodium complexes, *OrgLett* 2002) and the Schaub group (*JACS* 2018) showed already the feasibility of such process. Ruthenium-catalyzed alkylation of ammonia by alcohols was also reported (ref 22, work of the Vogt group), but, surprisingly, no reference from the group was mentioned (Beller *Angew. Chem. Int. Ed.*, 2010, 49, 8126 and 2011, 50, 7599). The authors also provide a reasonable plausible mechanism for the reaction based on some initial experimental observations.

Unfortunately, I do not feel that the work described in the present manuscript rises to the level of *Nature Communications*. Even if the results are good, this work looks like an improvement to previous ruthenium-chemistry.

Furthermore, the general quality of the manuscript (i.e. grammatical and typographical errors) detracts from the scientific merit of the data, surprising considering the quality of writing in many of the Beller group's previous papers, and a significant re-write to correct these errors is essential prior to publication in any journal.

Reply to the comments of the Reviewers

Reply to the comments of the Reviewer-1

Reviewer #1 (Remarks to the Author):

The research group of Beller and Jagadeesh have developed a synthetic method for the primary amines from carbonyl compounds using ammonia, molecular hydrogen and simple ruthenium precursor as a catalyst. Beller and other research group have reported Rh, Ir and including Ru based catalytic system for the similar transformation. However, the strength of this manuscript is widespread applicability and ardent display of substrate scope. In fact, such extensive investigations on functionalized and structurally diverse aldehydes and ketones leading to the highly useful primary amines are performed for the first time. Impressive scalability and standard mechanistic investigations have also been reported. This manuscript can certainly influence others in the field to develop strong substrate scope. This reviewer recommends publication of this work in Nature Communications after addressing following comments.

Reply: We are thankful to the Reviewer for highlighting the importance of our work and recommending its publication in Nature Communications.

1) Lines 77-79: redundant

Reply: We thank the reviewer for identifying the replication of the sentence. This has been corrected in the revised manuscript.

2) Lines 113: "design of catalyst" has to be changed as selection of catalyst (no new catalyst is designed for this work).

Reply: Thanks for the suggestion. In the revised manuscript we have changed it to "Selection of catalyst and reaction conditions for the reductive amination".

3) PPh₃

Reply: This discrepancy has been corrected throughout the revised manuscript.

4) Lines 117 and 125: typos

Reply: Thanks for identifying these unnoticed mistakes. All of these typographical errors have been corrected in the revised manuscript.

5) Table 2: desired conversion and yields have been attained in first two entries with no added ligands. Then, what is the need to have extensive ligands screening with poorly active catalyst in entries 3-12?

Reply: We agree with the comment of the Reviewer. The ligand screening using *in situ* generated catalysts was conducted to identify ligand effects in the reaction. With this approach, PPh₃, tris(4-

methoxyphenyl)phosphine and tris(*p*-tolyl)phosphine were identified to be the active ligands. We then tested the molecularly defined complex $\text{RuCl}_2(\text{PPh}_3)_3$ and found an increased activity. Based on the suggestion of Reviewer 2, we have prepared and tested the defined complexes $\text{RuCl}_2(\text{tris}(4\text{-methoxyphenyl)phosphine})_3$ (containing a ligand which was previously identified as active) and $\text{RuCl}_2(\text{tris}(4\text{-chlorophenyl)phosphine})_3$ (containing a ligand which was previously identified as inactive). This was done to test if the identified ligand trends also hold in molecularly defined complexes and to see if the improvement of activity for defined complexes can be generalized to other ligands. Indeed we observed similar activity for $\text{RuCl}_2(\text{tris}(4\text{-methoxyphenyl)phosphine})_3$ (95% benzylamine, see Table 2) compared to $\text{RuCl}_2(\text{PPh}_3)_3$ and saw a lower but relative to the *in situ* system increased activity for $\text{RuCl}_2(\text{tris}(4\text{-chlorophenyl)phosphine})_3$ (50% benzylamine, see Table 2). We have included these results in the text and changed the order of Table 2 (as per the suggestion of referee) and the result section to reflect this logical order for catalyst identification.

6) Figure 1: kinetic experiments clearly reveal that the homo-coupled imines as major intermediate (ca 60%). Thus, under the experimental conditions one would expect the presence of the corresponding hydrogenation product, secondary amine. Still no such product is observed (Table 2). Since the major intermediacy of internal imine is established in kinetics, a reaction pathway for the conversion of internal imines to primary amines needs to be considered in mechanistic proposal.

Reply: Thanks for raising this interesting point. Indeed, one can expect hydrogenation of the secondary imine to form the corresponding secondary amine. We did not observe this reaction under optimized conditions because of the following reasons: the secondary imine reacts with excess of ammonia to form equilibrium with benzylamine and the primary imine. From two separate hydrogenation experiments (see manuscript and SI) we have determined that the hydrogenation of both, the primary and the secondary imine can proceed under reaction conditions. In these experiments we performed the reaction of N-benzylidenebenzylamine under standard reaction conditions in presence and in absence of ammonia (see Fig S3 below). In presence of ammonia we observed >98% of benzylamine and traces of dibenzylamine. In absence of ammonia, however, N-benzylidenebenzylamine was completely hydrogenated to dibenzylamine. Also, at low pressures of ammonia (1-4 bar), we observed up to 20 % of dibenzylamine formation (please see new Fig. 1E). These results can be explained by a preferential hydrogenation of the primary imine over the secondary one. Due to the equilibrium induced by the ammonia pressure, the primary imine is replenished until all secondary imine is used up (resulting in quantitative conversion of the secondary imine to the primary amine). In the absence of ammonia or under reduced ammonia pressure, the replenishment of the primary imine is too slow, leading to hydrogenation of the secondary imine. We have added these results and the corresponding

discussion to the mechanistic section of the revised manuscript and we have expanded our proposed mechanism to include these additional reactions.

Fig. S3. Reaction of N-benzylidenebenzylamine in presence and absence of ammonia with RuCl₂(PPh₃)₃ and hydrogen.

7) Reconsider “for measuring NMR”.

Reply: This information has been changed to “Isolated as free amines and converted to hydrochloride salts. Corresponding hydrochloride salts were subjected to NMR analysis”. This info is now included in the revised manuscript wherever required.

8) Fig 2, compound 48 (49 in the revised manuscript): drawing error

Reply: We appreciate the reviewer critically going through our manuscript and rightly pointing out this error. This structural error has been corrected suitably in the revised manuscript.

9) Fig 6: diastereomeric ration shown for compound 82 but not 85.

Reply: The product 85 (86 in the revised manuscript) contains only one chiral center and hence there is no possibility of diastereomers for this product.

10) RuCl₂(PPh₃)₃ is described as both catalyst and pre-catalyst.

Reply: Thank you for pointing this out, since RuCl₂(PPh₃)₃ is only the precatalyst, this mistake has been corrected in the revised manuscript.

11) Lines 326-332: 31P {1H} NMR Spectrum

Reply: “NMR” has been added to all referenced spectra/measurements.

12) Fig 12: as the intermediacy of [RuHCl(PPh₃)₃] is established and it is readily available, reaction of it with independently prepared imine may allow prediction of more relevant intermediates.

Reply: As suggested by the Reviewer we have performed the reaction of secondary imine with $[\text{RuHCl}(\text{PPh}_3)_3]$ in presence of ammonia and hydrogen under standard reaction conditions. This reaction yields 30% of benzylamine and 69% of dibenzylamine. From this result it appears that at 2 mol% $[\text{RuHCl}(\text{PPh}_3)_3]$ is a potent hydrogenation (pre)catalyst, capable of quickly hydrogenation also the less reactive secondary imine. Under reaction conditions, $[\text{RuHCl}(\text{PPh}_3)_3]$ is likely present only in lower concentration, avoiding the direct hydrogenation of the secondary imine. We have added this experiment to the supporting information (Fig. S1).

Hydrogenation of N-benzylidenebenzylamine using $[\text{RuHCl}(\text{PPh}_3)_3]$.

13) Line 418: vials not vailes

Reply: Thanks for finding this mistake. We have now corrected it to vials.

14) Formation of metallic particles observed in NMR investigations. What about their involvement in reactions?

Reply: During the course of reaction in presence of substrate and ammonia, we did not observe the formation of metallic particles and hence we do not expect the involvement of a heterogeneous material in the reaction. However, the formation of particles was observed during NMR investigations presumably due to lower stability of *in situ* generated hydride species in absence of substrate and ammonia.

15) Impressive, in fact more attractive synthetic method for the primary amines is reported directly from primary (by Milstein group: Gunanathan, C.; Milstein, D. *Angew. Chem., Int. Ed.* 2008, 47, 8661) and secondary alcohols (by Beller group and others: Imm,S.; Baññ,S.; Neubert,L.; Neumann,H.; Beller,M. *Angew. Chem., Int. Ed.* 2010, 49, 8126 and Pinggen, D.; Müller, C.; Vogt, D. *Angew. Chem., Int. Ed.* 2010, 49, 8130) using ruthenium catalysts. As the carbonyl intermediacy is proposed in all such work, discussion of them in this manuscript is must (I am surprised by this omission).

Reply: We are highly thankful to the reviewer for this notable suggestion. In the introduction part we have now highlighted the preparation of primary amines from alcohols by including these references. Further, the reductive amination reaction has been studied with Ru-carbonyl catalysts and results have been discussed in the revised manuscript. Please go through these results below:

1. The $[\text{Ru}_3(\text{CO})_{12}] + \text{CataCXiumPCy}$ catalyst system reported by Imm,S.; Bähn,S.; Neubert,L.; Neumann,H.; Beller, M. *Angew. Chem., Int. Ed.* 2010, 49, 8126 and Pinggen, D.; Müller, C.; Vogt, D. *Angew. Chem., Int. Ed.* 2010, 49, 8130 has been tested for the

reductive amination of cyclohexanone (since cyclohexanol was the model substrate in those publications) under their reported reaction conditions. With this catalyst system we observed formation of 15 % primary amine.

Fig. S2. Reductive amination of cyclohexanone with ammonia using $\text{Ru}_3(\text{CO})_{12}/\text{CataCxiuPCy}$ catalyst system.

2. We have also tested $\text{RuHCl}(\text{CO})(\text{PPh}_3)_3$ which was used as pre-catalyst in Gallardo-Donaire, J.; Ernst, M.; Trapp, O.; Schaub, T.; *Adv. Synth. Catal.* 2016, 358, 358-363 (here $\text{RuHCl}(\text{CO})(\text{PPh}_3)_3$ and xantphos or dppe as a ligand were used), and as a precursor to obtain the catalyst system in Gunanathan, C.; Milstein, D. *Angew. Chem., Int. Ed.* 2008, 47, 8661. Furthermore, a related complex was also observed in our model *in situ* NMR system. With this complex we observed only 3% of benzylamine. Hence, in absence of more elaborate ligands, this carbonyl precursor appears to be less productive in the reductive amination reaction. We have added a discussion of this ligand effect in the manuscript. At this point it should be noted that in our original manuscript, we reported 40% benzylamine, 30% benzyl alcohol, 30% dibenzylamine for this reaction. During the revision of the manuscript, we have conducted this reaction three more times and in all cases found the below reported yields. We suspect that our previously used $\text{RuHCl}(\text{CO})(\text{PPh}_3)_3$ contained impurities which resulted in the erroneous yields. This change should not affect any of the conclusions in this work.

Fig. S2. Reductive amination of benzaldehyde using $\text{RuHCl}(\text{CO})(\text{PPh}_3)_3$

Once again we are extremely thankful to this reviewer for his/her constructive comments and suggestions, which enabled the improvement of our manuscript.

Reply to the comments of the Reviewer-2

Reviewer #2 (Remarks to the Author):

Summary of major claims of the paper: In this work, Jagadeesh, Beller, and co-workers show how a simple ruthenium based catalyst can effect the reductive amination of a variety of aldehydes and ketones. A number of first-row transition metals were screened, which showed the unique effectiveness of ruthenium. A screen of ligands showed triphenylphosphine was an effective ligand. A large substrate scope is presented. Aldehydes, including some very sterically hindered variants, are aminated in excellent yield. Halogens, a pinacolboronyl group, and a TMS protected alkyne are tolerated. Impressively, aldehydes with conjugated alkenes can be subject to reductive amination without concomitant reduction. At 3 mol% loading, ketones, including some fairly sterically hindered examples, can be aminated in good yield. A number of pharmaceutically relevant molecules were also aminated under these conditions. Seven examples were conducted on multigram scale, demonstrating the utility of this method. A mechanistic study suggests the intermediacy of ruthenium hydrides, and also suggests that ruthenium carbonyl complexes, which may be formed under the reaction conditions, are probably not catalytically relevant.

Reply: We are thankful to the reviewer for summarizing the highlights of our work.

Reviewer's opinion on merit of the work: While this is a nice piece of work, on a timely topic, I am not sure that as it stands that this paper will strongly influence thinking in the field. This work references two recently published papers, by the Schaub and Zhang groups (references 27 and 30), which conduct asymmetric reductive amination of ketones with hydrogen and ammonia or ammonium salts and chiral ruthenium complexes. These papers, which were published while this large body of work was presumably in progress, allow the asymmetric formation of amines with a lower loading of ruthenium than the present work. Nevertheless, this current work may have merit, since both of these papers use exotic and expensive ligands, whereas the current work uses simple and inexpensive ligands. In many cases, the resolution of racemic amines (through enzymatic resolution, or diastereomeric crystallization) is still competitive with their asymmetric synthesis, accordingly, a new synthesis of racemic amines, such as this work does have value.

Reply: We appreciate the reviewer for pointing out the value and prominence of our work.

The high pressures and temperatures required in this work probably make it more amenable to industrial use than small-scale transformations. Racemic amines are commonly prepared by the Leuckart-Wallach reaction, which does not require the use of transition metals, and is economical and scalable. Would the authors be able to present an analysis that the large scale reactions presented in this work are competitive with Leuckart-Wallach chemistry? Can the loading of

ruthenium be lowered? Are the amounts of ruthenium residue in the product below acceptable thresholds for pharmaceutical APIs?

Reply: We thank the Reviewer for his/her valuable suggestion to compare our work with the Leuckart-Wallach reaction. In this regard we tested 15 different aldehydes/ketones for the preparation of corresponding primary amines using the Leuckart-Wallach reaction. As can be seen from the table below this reaction works well for simple aldehydes/ketones and gave 53-75% yields of corresponding primary amines (Table S1; entries 1-3). For a majority of substituted and structurally diverse as well as heterocyclic aldehydes and ketones this method gave poor yields (5-15%) of the corresponding primary amines (Table S1; entries 4-9). For a few of the substrates this reaction did not work and the corresponding primary amines were not obtained (Table S1; entries 10-15). A majority of the sensitive functional groups were not tolerated. Gratifyingly, for all these substrates, the $\text{RuCl}_2(\text{PPh}_3)_3$ precatalyst using ammonia and hydrogen worked well and produced the corresponding primary amines in 72-93% yields. These results clearly reveal that catalytic reductive amination using $\text{RuCl}_2(\text{PPh}_3)_3$ is more generally applicable for the preparation of primary amines compared to the traditional Leuckart-Wallach reaction. These results and the table are included in the revised manuscript and supporting information (Table S1).

Table S1. Preparation of primary amines from aldehydes and ketones using Leuckart-Wallach reaction

Entry	Aldehyde /Ketone	Conv.	Yield of formamide (GC/GCMS yield)	Yield of primary amine (isolated yield)
1		>99	85% (10% diamino benzyl formamide) (3% tri benzylamine)	75%
2		75%	60% formamide 14% diamine	53% (14% diamine)
3		80%	75%	70%
4		>99%	90% (5-8% of corresponding acid)	5% (84% of benzyl amine. Boronic acid ester group was cleaved)
5		>99%	20% (78% of other products were observed)	15%
6		>99%	25% (20% pyridyl carboxylic acid. 53% other products)	15%

7		>99%	10% (88% quinoline carboxylic acid)	5%
8		>99%	30% (20% one C-C double bond cleaved formamide product. 48% of other products)	10%
9		30%	27%	5% (20% of de-iodo product)
10		>99%	0% (90% N-Benzyl-formamide. 9% benzoic acid. trimethylsilyl group was completely cleaved)	0% (88% benzylamine)
11		<2%	<1%	<1%
12		>99	<2% (95% N-Benzyl-formamide, cleaved product)	<1% (90% benzylamine)
13		<2%	>1%	-
14		>99%	0% (formamide was not observed. Other products were observed)	0%
15		<1%	<1%	<1%

Reaction conditions: 1-5 mmol of substrate, 5 equivalents of ammonium formate, 140 °C, 15-24 h. After the formation of corresponding formamide, 5 equivalents of aq.HCl (concentrated 37%) was added and refluxed at 110 °C for 9h, and filtered, washed with diethyl ether and then the filtrate was neutralized with NaOH. Finally, the product was extracted thoroughly with ethyl acetate to obtain the corresponding free primary amine.

As suggested by the reviewer we have tested a number of substrates using lowered catalyst loadings and the results are summarized below (Table S2 of SI). Unfortunately, all of the tested substrates (except for acetophenone) displayed lower activity at low catalyst loading (<2% for aldehydes and <3% for ketones). Thus, we could not obtain the desired primary amines in good

yields. Acetophenone worked at 1 mol % of catalyst loading and this result has been included in Scheme 5.

Table S2. Reductive amination of selected aldehydes and ketones using different $\text{RuCl}_2(\text{PPh}_3)_3$ loading

Entry	Aldehyde/ketone	$\text{RuCl}_2(\text{PPh}_3)_3$ loading (mol%)	Conversion	Product (s)
1		0.1	80%	0% Primary amine 77% Secondary imine
2		0.5	80%	19% Primary amine 60% Secondary imine
3		1	99%	70% Primary amine 25% Secondary imine 4% Side product 5
4		0.1	10%	9% secondary imine
5		0.5	80%	60% Primary amine 15% Alcohol 20% Secondary imine
6		1	99%	95% Primary amine 5% Alcohol
7		1	99%	99% Secondary imine
8		1	99%	99% Secondary imine
9		1	30%	30% Imine
10		1	30%	25% Primary amine
11		1	30%	20% Primary amine 5% Alcohol
12		1	80%	50% Primary amine 20% Alcohol 10% Secondary imine
13		1	75%	25% Primary amine 50% Alcohol

14		1	40%	20% Primary amine 20% Alcohol
15		1	70	65% Secondary imine
16		2	90%	25% Primary amine 65% Alcohol
17		2	88%	40% Primary amine 48% Alcohol
18		2	85%	70% Primary amine 15% Alcohol

Reaction conditions: 0.5 mmol benzaldehyde/ketone, 0.1-2 mol% $\text{RuCl}_2(\text{PPh}_3)_3$, 5-7 bar NH_3 , 40 bar H_2 1.5 mL t-amyl alcohol, 130 °C, 24 h

The isolated and purified benzylamine (by column chromatography), was subjected to Atomic Absorption Spectroscopy (AAS) analysis using aAnalytik Jena AG, Model: contrAA800D spectrophotometer to determine the Ru content. The measurement showed that no Ruthenium is present in the purified amine above the detection limit of 1 ppm. Please see the report below:

Dateiname 18062102						
Instrument: contrAA 800D # Tech: Flamme						
SW-Version: ASpect CS 2.2.1.0 Created: 21.06.2018 10:32						
Datum	Zeit	Name(2)	Linie	Konz.2	Einheit	Ext.
21.06.2018	10:18	Thiru TM 5	Ru349	Messwert 0.000	M- %	0.000
21.06.2018	10:19	Thiru TM 5	Ru349	Run.n.	M- %	0.000
AAS measurement for the detection of metal content in the isolated and purified amine product.						

While this work is presented as a way to prepare diverse families of amines on small scale, the formation and reduction of oxime ethers with borane (which should be referenced) is a very reliable way to prepare primary amines on small scale without the use of an autoclave.

Reply: According to the reviewer's suggestion we have mentioned this reaction in the introduction and suitable references have been included (references 46-48).

Specific questions:

Table 2 is somewhat uninformative. Entries 1 and 2, involving the use of preformed "molecularly defined" catalysts based on triphenylphosphine provided the best performance. All other phosphines were employed to generate the catalyst in-situ, with a ruthenium cymene complex as the ruthenium source. These in-situ generated catalysts all had inferior performance to the preformed catalysts, even in the case of triphenylphosphine. Accordingly, I feel in a ligand comparison, it would have made more sense to use preformed catalysts with some other phosphine ligands as well.

Reply: We do agree with the referee and appreciate this valuable suggestion. Accordingly, we have re-structured Table 2 and corresponding results in the revised manuscript. First, results of *in situ* generated catalysts have been given followed by the results of defined catalysts, reflecting the logical order for ligand identification and catalyst optimization. As suggested by you two defined catalysts, $(\text{RuCl}_2(\text{tris}(4\text{-methoxyphenyl})\text{phosphine})_3)$ and $\text{RuCl}_2(\text{tris}(4\text{-chlorophenyl})\text{phosphine})_3$, containing a more as well as a less active PPh_3 derivative, have been prepared and tested. $\text{RuCl}_2(\text{tris}(4\text{-methoxyphenyl})\text{phosphine})_3$ was found to display similar activity compared to $\text{RuCl}_2(\text{PPh}_3)_3$ (95% of benzyl amine, see SI), while $\text{RuCl}_2(\text{tris}(4\text{-methoxyphenyl})\text{phosphine})_3$ is less active compared to $\text{RuCl}_2(\text{PPh}_3)_3$ (50 % of benzylamine), but more active compared to the *in situ* system (10% benzylamine, see Table 2 entry 6). Therefore, the defined complexes reflect the same ligand trend observed in case of *in situ* generated catalysts. The following modified table is included in the revised manuscript:

Table 2. Reductive amination of benzaldehyde using *in situ* generated and molecularly defined ruthenium catalysts.

Entry	Ru-precursor/ Defined Ru-catalyst	L	Yield (%)			
			2	3	4	5
1 ^a	$[\text{RuCl}_2(\text{p-cymene})]_2$	L1	40	2	40	17
2 ^a	$[\text{RuCl}_2(\text{p-cymene})]_2$	L2	5	-	60	33
3 ^a	$[\text{RuCl}_2(\text{p-cymene})]_2$	L3	2	-	25	70
4 ^a	$[\text{RuCl}_2(\text{p-cymene})]_2$	L4	50	5	20	24

5 ^a	[RuCl ₂ (p-cymene)] ₂	L5	53	4	16	25
6 ^a	[RuCl ₂ (p-cymene)] ₂	L6	10	2	42	44
7 ^a	[RuCl ₂ (p-cymene)] ₂	L7	-	-	20	76
8 ^a	[RuCl ₂ (p-cymene)] ₂	L8	-	-	18	79
9 ^a	[RuCl ₂ (p-cymene)] ₂	L9	-	-	25	74
10 ^a	[RuCl ₂ (p-cymene)] ₂	L10	-	-	30	68
11 ^b	RuCl ₂ (PPh ₃) ₃	-	95	4	-	-
12 ^b	RuCl ₂ (PPh ₃) ₄	-	92	7	-	-
13	RuCl ₂ (tris(4-methoxyphenyl)phosphine) ₃	-	95	4	-	-
14	RuCl ₂ (tris(4-chlorophenyl)phosphine) ₃	-	50	-	49	-

Reaction conditions: ^a0.5 mmol benzaldehyde, 1 mol% [RuCl₂(p-cymene)]₂ (2 mol% corresponds to monomer), 6 mol% ligand, 5-7 bar NH₃, 40 bar H₂, 1.5 mL *t*-amyl alcohol, 130 °C, 24 h, GC yields using n-hexadecane as standard. ^bSame as 'a' using, 2 mol% defined catalyst.

Kinetic analysis shows the presence of side products. I would like to see further commentary and analysis regarding the formation and disappearance of compound 5. Is this compound, which must arise from a series of redox reactions, including a C-C bond formation, turned into product, or polymeric material after time?

Reply: Williams *et al.* (see new Ref. 78 manuscript) and Corey *et al.* (see new Ref. 79 manuscript) have reported the formation of **5** via initial condensation of three equivalents of primary imine to form intermediate **98** (see updated Fig. 12 in the revised manuscript), followed by thermal cyclization to yield **5**. We expect that under reaction conditions, the facile hydrogenation of the primary imine (which is in equilibrium with the secondary imine) keeps the stationary concentration low, avoiding the necessary trimerization to form **99** and then **5**. When the hydrogenation conditions are not optimal, larger amounts of the primary imine can be present, leading to the formation of **5**. Correspondingly, reaction of a mixture of benzaldehyde and secondary imine **4**

under standard reaction conditions except for the absence of H₂ leads to formation of **5** in 20% yield (see new Fig. S4). Furthermore, as can be seen in Fig. 1D, the hydrogenation activity of the catalyst system drops rapidly for temperatures > 130 °C. Therefore, we expect that while the reaction is cooled down, the hydrogenation reactions quickly ceases, leading to a higher concentration of primary imine. While the thermal energy is still sufficient, the primary imine can then trimerize (which proceeds even at room temperature, see Ref. 78 manuscript) and cyclize (which requires elevated temperature) to form **5**. While this side reaction during cool down would not affect the optimized reaction (since there is no primary imine left after complete hydrogenation), it could occur for incomplete reactions (e.g. when the reaction is stopped prematurely). In line with this reasoning, we found that when the optimized benzaldehyde amination is stopped after just 30 min, 30% yield for **5** is obtained (see Fig. S4). Therefore, we propose that formation and apparent decrease of **5** in the concentration-time profile is an artifact resulting from prematurely quenching the reaction.

In a separate reaction, we exposed **5** to our standard amination reaction conditions and found no conversion. Therefore, **5** cannot be converted to the desired product (see Fig. S4).

A discussion of the formation of **5** has been included in the mechanistic section and Fig. 12 has been extended to include the reaction pathway for this transformation. The corresponding experiments have been added to the supporting information.

Fig. S4. Control reactions to confirm the formation of side product **5**.

Once again we are highly grateful to this reviewer for his/her constructive comments and valuable suggestions, which enabled the improvement of our manuscript.

Reply to the comments of the Reviewer-3

Reviewer #3 (Remarks to the Author):

In the current manuscript, Beller and Jagadeesh describe a ruthenium-catalyzed reductive amination. As such this process is not new, but it is more general. Previous reports from the Beller group (using rhodium complexes, *OrgLett* 2002) and the Schaub group (*JACS* 2018) showed already the feasibility of such process. Ruthenium-catalyzed alkylation of ammonia by alcohols was also reported (ref 22, work of the Vogt group), but, surprisingly, no reference from the group was mentioned (Beller *Angew. Chem. Int. Ed.*, 2010, 49, 8126 and 2011, 50, 7599). The authors also provide a reasonable plausible mechanism for the reaction based on some initial experimental observations. Unfortunately, I do not feel that the work described in the present manuscript rises to the level of *Nature Communications*. Even if the results are good, this work looks like an improvement to previous ruthenium-chemistry. Furthermore, the general quality of the manuscript (i.e. grammatical and typographical errors) detracts from the scientific merit of the data, surprising considering the quality of writing in many of the Beller group's previous papers, and a significant re-write to correct these errors is essential prior to publication in any journal.

Reply: We thank the reviewer for this assessment of our manuscript. In response to the concerns raised by the reviewer with regards to the importance of our results, we would like to discuss some of the referenced publications, illustrating the relevance of $\text{RuCl}_2(\text{PPh}_3)_3$ catalyzed reductive amination.

1. We agree that the reported Rhodium catalyst (*Org. Lett.* 2002, 4, 2055-2058) represents an important breakthrough in homogeneous reductive amination catalysis. The main drawback of this work is that the catalyst system is applicable only for the reductive amination of a few aldehydes and lacks the ability to aminate more complex aldehydes or ketones. In addition, utilization of Rhodium presents economical drawbacks compared to Ruthenium.
2. Thanks for drawing attention to the important work of Schaub (*J. Am. Chm. Soc.* 2018, 140, 355-361), which reports the asymmetric reductive amination of ketones for the preparation of chiral branched amines. We absolutely agree that this work is of potential interest for reductive amination processes. It should be noted that for the preparation of chiral amines, the use of Schaub's catalyst system ($\text{RuHCl}(\text{CO})(\text{PPh}_3)_3$ -*(S,S)*-f-binaphane) can be an excellent approach, but the synthesis of racemic or achiral amines call for a broadly applicable, inexpensive, achiral catalyst. Also, for the preparation of chiral amines, resolution of racemic amines (through enzymatic resolution, chiral resolution or diastereomeric crystallization) is often competitive with asymmetric syntheses. Furthermore, the reported catalyst is only applied for simple ketones and its applicability was not demonstrated for the general synthesis of functionalized, structurally challenging or complex primary amines. An economically

important principle is that to achieve a convenient and practical chemical synthesis, the catalyst must be simple, effective and commercially available and/or easily accessible. In this regard, our methodology using simple, easily available and broadly applicable $\text{RuCl}_2(\text{PPh}_3)_3$ clearly fulfills this requirement and hence represents an important development in reductive amination catalysis. To the best of our knowledge, until today, there is no homogeneous catalyst system that has achieved the preparation of a diverse range of primary benzylic, heterocyclic and aliphatic amines. In fact, such extensive investigations on functionalized and structurally diverse aldehydes and ketones leading to highly useful primary amines are performed for the first time.

3. Apart from the merit of preparing such a diverse range of primary amines, our work also contributes to the mechanistic understanding of ruthenium catalyzed reductive amination. Through detailed kinetic and *in situ* spectroscopic investigations, we have clarified the formation of intermediates and side products in the reaction and provided deeper insight into catalyst activation as well as deactivation pathways. This understanding will hopefully find wide-spread use in future reductive amination and hydrogenation research.
4. We have cited and highlighted the works related to the Ruthenium-catalyzed alkylation of ammonia by alcohols (references 39-42).
5. Thanks for bringing attention to grammatical and typographical errors. We have completely revised the manuscript to address this concern.

Based on the above presented merits, we strongly believe that our work is important and suitable for the publication in Nature Communications. We hope that the reviewer will reconsider his/her opinion and agree to the publication of our work.

Reviewer #1 (Remarks to the Author):

The authors have appropriately addressed all concerns pointed out. This report provides an important and simple synthetic tool for preparing diverse primary amines. Thus, I recommend publication in its current form in Nature Communications.

Chidambaram Gunananthan

Reviewer #2 (Remarks to the Author):

I have read the revised manuscript, and feel that the authors have addressed all concerns that I raised during my initial review. The context and value of this reaction is better explained. I especially appreciate the time that must have been taken to perform comparative experiments, and I feel they certainly went above and beyond what I expected. The rebuttal work was obviously done very carefully and thoroughly, and has increased the value of the paper.

This paper represents a significant practical advance in the synthesis of primary amines, and should also serve as a leading reference that summarizes what the state of this field is. In addition, the thorough analysis of side products will be very informative to practitioners of the field. I now support the publication of this revised paper in Nature Communications.

Reviewer #3 (Remarks to the Author):

In the last years, many papers, including the authors' previous papers for the reductive amination, have reported Ru-catalyzed hydrogenation/dehydrogenation reactions and related reactions for C-C and C-N bonds formation. Compared with the reported examples, the present reaction is just broader. Because of lack of novelty and scientific impact, this reviewer does not recommend the manuscript for publication in this journal.

Reply to the comments of the Reviewers

Reviewer #1 (Remarks to the Author):

The authors have appropriately addressed all concerns pointed out. This report provides an important and simple synthetic tool for preparing diverse primary amines. Thus, I recommend publication in its current form in Nature Communications.

Reply: We are extremely thankful to the reviewer for recommending our work to be published in Nature Communications.

Reviewer #2 (Remarks to the Author):

I have read the revised manuscript, and feel that the authors have addressed all concerns that I raised during my initial review. The context and value of this reaction is better explained. I especially appreciate the time that must have been taken to perform comparative experiments, and I feel they certainly went above and beyond what I expected. The rebuttal work was obviously done very carefully and thoroughly, and has increased the value of the paper. This paper represents a significant practical advance in the synthesis of primary amines, and should also serve as a leading reference that summarizes what the state of this field is. In addition, the thorough analysis of side products will be very informative to practitioners of the field. I now support the publication of this revised paper in Nature Communications.

Reply: We are highly grateful to the reviewer for his/her remarks on our revision and recommendation to accept our work to be published in Nature Communications.

Reviewer #3 (Remarks to the Author):

In the last years, many papers, including the authors' previous papers for the reductive amination, have reported Ru-catalyzed hydrogenation/dehydrogenation reactions and related reactions for C-C and C-N bonds formation. Compared with the reported examples, the present reaction is just broader. Because of lack of novelty and scientific impact, this reviewer does not recommend the manuscript for publication in this journal.

Reply: We strongly believe that our work is of potential interest and impact on the synthesis of functionalized and structurally diverse primary amines using a simple and commercially available Ru-catalyst.